# Low SNR Multi-Emitter Signal Sorting and Recognition Method Based on Low-Order Cyclic Statistics CWD Time-Frequency Images and the YOLOv5 Deep Learning Model

**DOI:** 10.3390/s22207783

**Published:** 2022-10-13

**Authors:** Dingkun Huang, Xiaopeng Yan, Xinhong Hao, Jian Dai, Xinwei Wang

**Affiliations:** Technology on Electromechanical Dynamic Control Laboratory, School of Mechatronical Engineering, Beijing Institute of Technology, Beijing 100081, China

**Keywords:** cyclic stationary analysis, CWD time-frequency analysis, noise suppression, YOLOv5 model, radiation source signal sorting and identification

## Abstract

It is difficult for traditional signal-recognition methods to effectively classify and identify multiple emitter signals in a low SNR environment. This paper proposes a multi-emitter signal-feature-sorting and recognition method based on low-order cyclic statistics CWD time-frequency images and the YOLOv5 deep network model, which can quickly dissociate, label, and sort the multi-emitter signal features in the time-frequency domain under a low SNR environment. First, the denoised signal is extracted based on the low-order cyclic statistics of the typical modulation types of radiation source signals. Second, the time-frequency graph of multisource signals was obtained through CWD time-frequency analysis. The cyclic frequency was controlled to balance the noise suppression effect and operation time to achieve noise suppression of multisource signals at a low SNR. Finally, the YOLOv5s deep network model is used as a classifier to sort and identify the received signals from multiple radiation sources. The method proposed in this paper has high real-time performance. It can identify the radiation source signals of different modulation types with high accuracy under the condition of a low SNR.

## 1. Introduction

A radiation source signal has the characteristics of variable parameters, various modulation forms, and comprehensive space coverage. In order to expand the function and anti-interference ability of a radiation source signal, the modulation mode is also developing. The modulation mode has gradually changed from a continuous wave system to multi-waveform frequency modulation, pseudo-random-code phase modulation, frequency-agile, pulse Doppler, and other complex modulation modes. In the actual scene, the measured signal is often mixed with other signals and electromagnetic noise, which will reduce the recognition and analysis ability of the receiver. This phenomenon is more evident in an electromagnetic environment with low SNR. Because the signals of various radiation sources are widely present, the communication equipment of manufactured equipment is likely to be interfered with by them in extreme environments. Therefore, successfully identifying the modulation type of the mixed radiation source signal is essential to ensure communication-equipment stability. After identifying the modulation type of the radiation source signal, the equipment can estimate its detailed parameters and avoid interference. The accurate identification of modulation types of mixed emitter signals is an important signal-processing-system index and one of the core technical problems of blind signal adaptive processing.

The difficulties of emitter signal sorting and identification under a low SNR mainly include the following three points:

(1) The noise-suppression effect of a signal and the contradiction of real-time processing

In an electromagnetic environment with low SNR, the receiver must accumulate signal energy for a long time to achieve a good noise-suppression effect. This method increases the computation amount of the signal-analysis algorithm and reduces the real-time performance. The large working bandwidth disperses the FM signal’s energy, further reducing the identification ability, and the effect of the ordinary noise-suppression method is poor.

(2) There are many factors affecting the recognition accuracy of artificial neural networks. Under the same preprocessing effect, the classifier can only obtain the recognition result of a single category and cannot obtain the location information. This is not conducive to the subsequent parameter-estimation work. It is necessary to use large-scale artificial neural network architecture to obtain high recognition accuracy and a low missed detection rate, which extends the training time.

(3) It is difficult to accurately identify the mixed signal combination mode in the coarse classification work.

The sub-signal features in the mixed radiation source are easily confused, which will mislead the classifier and produce incorrect classification results. An inefficient modulation feature-recognition method will significantly reduce the recognition accuracy of the classifier in the working scene.

To solve the problem of radiation source-signal sorting and identification under low SNR, [1] calculates the second-order cyclic cumulant and fuses two detectors to increase the types of signal recognition and reduce the computational burden. However, this method has a low detection probability for signals below −10 dB. The authors of [2] calculate the first-order cyclic mean of a VHF signal, identify the modulation classification mode, and obtain a high recognition rate of the blind signal modulation mode, but they identify fewer signal types. In [3], second-order cyclic autocorrelation function and a convolutional neural network are used to compress the signal dimension and reduce the computation amount, and the recognition accuracy is high. The authors of [4] used a multilevel LPI-NET classifier combined with CWD time-frequency analysis. The scheme can recognize 13 radar waveforms, and the recognition accuracy of the 0 dB signal-to-noise ratio is more than 98%. The authors of [5] study generalized Hankel matrix factorization and propose a new cyclic mean kernel function, which can improve the accuracy of parameter estimation in a low SNR electromagnetic environment. The method proposed in [6] accumulates signal energy and suppresses noise, and uses a deep confidence network to automatically sort and identify radiation source signals. The accuracy of this method is 90% at −5 dB. The method in [7] integrates multidimensional information of CWD, STFT, and cyclic spectrum, adopts three kinds of neural networks and flying fish swarm intelligence algorithms to identify the modulation types of signals, and the recognition rate reaches 84.7% at −4 dB. The authors of [8] studied the autocorrelation function of the signal and used CNN-DNN to identify the two-dimensional cross-section features, with a high recognition accuracy at −5 dB. The method proposed in [9] uses the DCN-BILSTM bidirectional long short-term memory network to recognize the amplitude and phase information of signals, and the recognition rate is higher when the SNR is 4 dB. The authors of [10] propose RRSARNet, a new artificial neural network based on meta-transfer learning, which can recognize STFT images and adaptively recognize signals. In [11], to solve the problem of LPI radar waveform recognition at low SNR, a triple convolutional neural network was designed to recognize PWVD time-frequency images, and the recognition accuracy reached 94% at −10 dB. In [12], the CMWT time-frequency graph was input into the ResNet (SEP-ResNet) network, which could classify the enhanced T-F images and complete the high-precision intelligent recognition of radar-modulated signals in a low SNR environment. When the SNR is −10 dB, the recognition accuracy is 93.44%. The authors of [13] combined time-frequency analysis with a h deep neural network and analyzed the distribution law of electromagnetic noise in the time-frequency diagram, designed a filter to suppress the noise, and obtained a good classification effect. The method in [14] combines CWD and CNN neural networks, uses the binarization operation of preprocessed time-frequency images to suppress noise and highlight signal features, and the signal-recognition accuracy exceeds 90% at −4 dB, and the recognition accuracy of some signals exceeds 98%. The authors of [15] adjust the hyperparameters and filters of different signals in the CNN network and control the calculation amount through sample averaging, with the detection accuracy exceeding 80% at −15 dB. The method proposed in [16] combines Transformer and YOLOv5 for UAV target detection and obtains a good AP value. The method proposed in [17] adopts deep convolutional network and K-means clustering, and a YOLOv5 classifier to achieve high real-time small-target-object detection. In [18], CAA-Yolo was used to identify ship targets on the sea surface, highlighting the recognition of small targets and increasing the mAP value by 3.4%. The authors of [19] combined trunk learning and a Yolo-Tiny network, transplanted the model to GPU edge devices for use, and obtained a high recognition rate. On the basis of Yolo-X, ref. [20] combines the adaptive activation function Meta-ACon with the SPP module of Backbone to improve the feature extraction ability. The authors of [21] introduce a real-time multi-class target recognition system based on millimeter wave (mmWave) imaging radar for advanced driver assistance systems. Objects in various scenes can be detected with little change in the training dataset, thus reducing the training cost. In [22], radar information is processed by mapping transform neural networks to obtain mask mapping, so that radar information and visual information are on the same scale. This paper proposes a multi-data source deep learning Object Detection network (MS-Yolo) based on millimeter wave radar and vision fusion. The authors of [23,24,25,26,27] used different deep learning models and preprocessing algorithms to classify and recognize radar signals, ships, and intracellular molecules. In [28,29,30], generative adversarial network and adaptive noise suppression are used to sort and identify modulation features for radar signals, and good results are obtained.

As shown in Figure 1, the technical route of spectrum analysis, time-frequency analysis, and other preprocessing methods combined with an artificial neural network classifier has become the primary solution to the problem of emitter-signal sorting and identification under low SNR. However, the real-time performance of the neural network in identifying long sequence signals is poor. Through the above research, this paper finds that the main solutions have high signal classification accuracy from 0 dB to −10 dB, while sorting and identification accuracy below −10 dB is significantly reduced. Spectral analysis, time-frequency analysis, and other methods have a poor effect on signal processing, which brings difficulties to sorting artificial neural networks. In general, real-time performance is negatively correlated with the effect of preprocessing.

To solve the difficult problem of radiation source-signal sorting and identification under a low SNR, this paper performs the following research work and innovation, as illustrated in Figure 2.

Low-order cyclic statistics and CWD time-frequency analysis are used as the preprocessing methods for received signals. Using the granularity of the cycle frequency to adjust the resolution of the cycle mean, combined with the IFFT algorithm, can suppress the noise, reduce the amount of computation, and improve the real-time performance of the preprocessing algorithm.To mark and identify the modulation features of multiple signals on the time-frequency graph, we adopted the YOLOv5 deep neural network framework as the classifier. After training, the single recognition time is short, and multiple sub-signal features can be recognized simultaneously. The output result of the classifier is the basic premise of parameter estimation.To reduce the missed detection rate and identify the combination pattern of mixed signals, the labeled and unlabeled datasets are fused to train the classifier. Using different kinds and quantities of signals in the test can show the recognition defects of the classifier at different levels and support the subsequent optimization work.

## 2. Materials and Methods

### 2.1. Multisource Signal Preprocessing Method Based on Cyclic Mean and CWD Time-Frequency Analysis

The preprocessing method proposed in this paper takes the cyclic mean calculation as the denoising method, carries out domain transformation through the IFFT, and finally generates a time-frequency image, including the multi-emitter signal, through CWD time-frequency analysis. The length and domain of the sequence will change during each step. The input to the preprocessing system is a long one-dimensional-sequence discrete signal with noise in the time domain. The output is a two-dimensional CWD time-frequency image after noise suppression in the short sequence.

Since the specific order statistical characteristic parameters of cyclically stationary signals change periodically with time and the characteristics of different classes of modulated signals are distinguished, low-order statistics require less computation and have good noise-suppression performance. The cyclic mean is applied as the first step of the preprocessing algorithm. Since most received signals have cyclic stationary characteristics, the cyclic mean values of AM and FM modulation types are theoretically derived in this paper.

#### 2.1.1. Cyclic Mean Analysis of AM Signals

As a common modulation signal type, AM signals are widely used in the field of digital communication. The mathematical model is:(1)$$u_{AM}(t)=U_{cm}(1+m_{a}\cos(f_{m}t))\cos(2\pi f_{c}t)+n(t)$$

In the above equation, $U_{cm}$ is the carrier signal amplitude, $m_{a}=K(U_{\Omega m}/U_{cm})$ is the modulation rate, K is a constant, $U_{\Omega m}$ is the modulated signal amplitude, $f_{m}$ is the modulated signal frequency, $f_{c}$ is the carrier frequency, and n(t) is the Gaussian white noise. The cyclic mean of the AM signal when the cyclic frequency is α is:(2)$$M_{u}^{\alpha }(t)=<U_{cm}(1+m_{a}\cos(f_{m}t))\cos(2\pi f_{c}t)e^{-j2\pi \alpha t}>+<n(t)e^{-j2\pi \alpha t}>$$

Since the mean of the Gaussian white noise is 0, the value of $<n(t)e^{-j2\pi \alpha t}>$ is 0, from which we can prove that the first-order cyclic mean has theoretically good denoising properties. Expanding (1) and (2) to obtain the sum of the two parts, part A is:(3)$$M_{uA}^{\alpha }(t)=<U_{cm}\cos(2\pi f_{c}t)[\cos(2\pi \alpha t)-j\sin(2\pi \alpha t)]>$$

Expanded, it is:(4)$$\begin{array}{l} M_{uA}^{\alpha }(t)=<\frac{1}{2}U_{cm}\{[\cos(2\pi (f_{c}+\alpha )t)]+[\cos(2\pi (f_{c}-\alpha )t)]\}\\ -j\frac{1}{2}U_{cm}\{[\sin(2\pi (f_{c}+\alpha )t)]+[\sin(2\pi (f_{c}-\alpha )t)]\}> \end{array}$$

Part B is:(5)$$M_{uB}^{\alpha }(t)=<U_{cm}m_{a}\cos(2\pi f_{c}t)\cos(f_{m}t)[\cos(2\pi \alpha t)-j\sin(2\pi \alpha t)]>$$

Expanded, it is:(6)$$M_{uB}^{\alpha }(t)=<\frac{1}{2}m_{a}U_{cm}[\cos(2\pi f_{c}t+f_{m}t)+\cos(2\pi f_{c}t-f_{m}t)][\cos(2\pi \alpha t)-j\sin(2\pi \alpha t)]>$$

$R=0.5*m_{a}U_{cm}$, the expansion of AM signal part B can be decomposed into C, D, E, and F:(7)$$M_{BC}^{\alpha }=<R\{\frac{1}{2}[\cos(2\pi t(f_{c}+\frac{1}{2\pi }f_{m}+\alpha ))+\cos(2\pi t(f_{c}+\frac{1}{2\pi }f_{m}-\alpha ))]>\;M_{BD}^{\alpha }=<-jR\{\frac{1}{2}[\sin(2\pi t(f_{c}+\frac{1}{2\pi }f_{m}+\alpha ))-\sin(2\pi t(f_{c}+\frac{1}{2\pi }f_{m}-\alpha ))]>\;M_{BE}^{\alpha }=<R\{\frac{1}{2}[\cos(2\pi t(f_{c}-\frac{1}{2\pi }f_{m}+\alpha ))+\cos(2\pi t(f_{c}-\frac{1}{2\pi }f_{m}-\alpha ))]>\;M_{BF}^{\alpha }=<-jR\{\frac{1}{2}[\sin(2\pi t(f_{c}-\frac{1}{2\pi }f_{m}+\alpha ))-\sin(2\pi t(f_{c}-\frac{1}{2\pi }f_{m}-\alpha ))]>$$

From Equations (3)–(7), the cyclic mean of the AM signal is divided into real and imaginary parts, and a parameter with cyclic frequency α is added to the Fourier frequency. Because the frequency terms of the two parts are the same, this section only uses the real part mean as an example. The real part of the cyclic mean is composed of three spectral lines with different frequencies, where the cyclic frequency α represents the frequency under the current transformation. The image of the first-order cyclic mean is the result of time averaging after the Fourier transform of the signal is shifted α to the left as a whole.

Figure 3 shows the cyclic mean of the AM signal under different SNRs. In the range from 0 to −10 dB, the cyclic mean of the signal has a substantial noise-suppression property, and at −20 dB, the noise-suppression performance is significantly reduced.

#### 2.1.2. Cyclic Mean Analysis of the FM Signal

The mathematical model of an FM signal is:(8)$$u_{fm}(t)=U_{m}\cos(2\pi f_{c}t+m_{f}\sin\Omega t)+n(t)$$

For the carrier-signal amplitude $U_{m}$, carrier frequency $f_{c}$, Gaussian white noise n(t), modulation frequency Ω, $m_{f}=\frac{\Delta \omega }{\Omega }$ and frequency bias Δω, the cyclic mean of the FM signal when the cyclic frequency is α is:(9)$$M_{u}^{\alpha }(t)=<u_{fm}(t)e^{-j2\pi \alpha t}>=<U_{m}(\cos(2\pi f_{c}t+m_{f}\sin\Omega t)e^{-j2\pi \alpha t})>+<n(t)e^{-j2\pi \alpha t}>$$

Similarly to an AM signal, the mean value of white Gaussian noise is zero. The following can be obtained:(10)$$M_{ufm}^{\alpha }(t)=<u_{fm}(t)e^{-j2\pi \alpha t}>=<U_{m}(\cos(2\pi f_{c}t+m_{f}\sin\Omega t)e^{-j2\pi \alpha t})>$$

By expanding Equation (10), the following can be obtained:(11)$$M_{ufm}^{\alpha }(t)=<U_{m}(\cos(2\pi f_{c}t+m_{f}\sin\Omega t)(\cos(2\pi \alpha t)-\mathrm{jsin}(2\pi \alpha t)>$$

Expanding Equation (11) into real and imaginary parts:(12)$$M_{ufmA}^{\alpha }(t)=<U_{m}(\cos(2\pi f_{c}t+m_{f}\sin\Omega t)(\cos(2\pi \alpha t)>$$
(13)$$M_{ufmB}^{\alpha }(t)=<U_{m}(\cos(2\pi f_{c}t+m_{f}\sin\Omega t)(\mathrm{jsin}(2\pi \alpha t)>$$

Expanding (11) and (12):(14)$$M_{ufmA}^{\alpha }(t)=<\frac{1}{2}U_{m}[\cos(2\pi t(f_{c}+\alpha +\frac{\Delta \omega }{2\pi }\sin c(\Omega t))+\cos(2\pi t(f_{c}-\alpha +\frac{\Delta \omega }{2\pi }\sin c(\Omega t))]>$$
(15)$$M_{ufmB}^{\alpha }(t)=<-\frac{j}{2}U_{m}[\sin(2\pi t(f_{c}+\alpha +\frac{\Delta \omega }{2\pi }\sin c(\Omega t))-\sin(2\pi t(f_{c}-\alpha +\frac{\Delta \omega }{2\pi }\sin c(\Omega t))]>$$

According to Equations (14) and (15), the cyclic mean of the FM signal can also be divided into two parts: the real part and the imaginary part. Similar to the result after the Fourier transform of the FM signal, a component of cyclic frequency is added to the spectral line.

As shown in Figure 4, when the cyclic frequency α is close to the harmonic frequency of the signal, the cyclic mean will have a peak value, and when it leaves the harmonic frequency, the peak value will rapidly decrease to zero. As the calculation method of the cyclic mean is the generalized FFT operation, due to its superior denoising performance and high operation speed, the calculation of the cyclic mean is the main means of denoising and preprocessing the received signals from multiple sources. Because the traditional method of lengthy signal spectrum analysis and time-frequency analysis directly transforming domain operations tends to increase the amount of calculation and consume too much time, choosing an appropriate cyclic frequency is equal to the adjusted cyclic average resolution. With its sequence control points, the average cyclic images can carry long-time domain-sequence information with short sequences and reduce the time for subsequent steps.

### 2.2. CWD Time-Frequency Analysis Image Generation

The time-domain sequence is converted into the cyclic domain by calculating the cyclic mean of the multi-emitter signal, and the cyclic frequency controls the sequence length. Since the input to the CWD algorithm is a time-domain sequence, after the cyclic mean suppresses the noise, the IFFT method is used to transform the short sequence in the cyclic domain into a time-domain sequence of the same length. Because the value of the cyclic frequency is small, the frequency offset caused by the IFFT can be ignored. The above steps can maintain the noise-suppression effect, complete the sequence length conversion, and reduce the processing time of the CWD algorithm.

The kernel function determines the type of time-frequency analysis. This paper uses the unified expression of Cohen class time-frequency distribution to derive the theory of CWD time-frequency analysis. Choosing the proper kernel function can reduce the cross-term caused by the operation, if:(16)$$\varphi (\tau ,\xi )=e^{-\xi^{2}\tau^{2}/\sigma }$$

Among these, σ(σ >0) is known as the scaling factor, which substitutes (1–11) into the unified expression of the Cohen class time-frequency distribution. If the continuous signal is x(t), then the CWD is expressed as:(17)$$CWD_{x}(t,\omega )=\int_{-\infty }^{\infty }e^{-j\omega t}[\int_{-\infty }^{\infty }\sqrt{\frac{\sigma }{4\pi \tau^{2}}}G(\mu ,\tau )z(\mu +\frac{\tau }{2})z*(\mu -\frac{\tau }{2})d\mu ]d\tau$$
where
(18)$$G(\mu ,\tau )=e^{-\sigma {\left(\mu -t\right)}^{2}/(4\tau^{2})}$$

At time t, the angular frequency is ω and σ is the scaling factor. The size of the scaling factor is proportional to the amplitude of the crossover term. Therefore, it is crucial for the choice of the CWD scaling factor; a small scale factor will lead to poor time-frequency aggregation of the signal, and a large scaling factor will lead to a serious cross-term.

In practice, the CWD calculation needs to be discretized by:(19)$$CWD_{s}\left(s,\omega \right)=2\overset{\infty }{\underset{\tau =-\infty }{\sum }}W_{N}\left(\tau \right)e^{-j2\omega \tau }\overset{\infty }{\underset{\mu =-\infty }{\sum }}W_{M}\left(\mu \right)\sqrt{\frac{\sigma }{4\pi \tau^{2}}}e^{-\sigma {\left(\mu -l\right)}^{2}/\left(4\tau^{2}\right)}z\left(\mu +\tau +s\right)z*\left(\mu -\tau +s\right)$$

In Equations (1)–(19), the window functions $W_{N}(\tau )$ and $W_{M}(\tau )$ are weighted. The τ values of these two window functions are between -n/2≤τ≤N/2 and -m/2≤τ≤M/2, respectively.

CWD, as a widespread use of the time-frequency analysis method, has an excellent cross-term inhibiting effect, but it is rather time-consuming for the two-dimensional sequence of the process. To improve the speed of pretreatment, one-dimensional time series are directly processed. For a one-dimensional sequence on noise-suppression generated CWD time-frequency images, the time-frequency transform after image processing minimizes time consumption, and there is no need to operate the two-dimensional image matrix frequently. When CWD time-frequency images are generated, they can be directly fed into the classifier, which greatly improves the processing speed.

### 2.3. Comparison of the Results of the Preprocessing Algorithm

Multi-emitter signal superposition can be divided into two cases. The first case is that the time-frequency domain of each sub-signal is far apart, and a method of searching for the peak value in the frequency domain can effectively distinguish the modulation type of the sub-signal. The second case is that each sub-signal overlaps in the time-frequency domain. This study focuses on the second case.

Figure 5 shows the cyclic mean of AM and FM signals mixed at −20 dB. After noise suppression using the cyclic mean, the characteristics of each sub-signal can be displayed. The cyclic mean value of the received mixed signals is calculated, and the number of sampling points is added to the time-domain sequence to achieve a better noise-suppression effect. The IFFT operation is then performed on the signal to obtain the time-domain signal under short sequence points. At this time, the algorithm has completed the transformation of long and short time series.

Figure 6 and Figure 7 show the CWD time-frequency images after noise suppression at −20 dB and −15 dB, respectively. The preprocessing method proposed in this paper has a good noise-suppression effect on different combinations of sub-signals. However, the CWD transformation diagram at −20 dB still retains certain noise points, while the signal time-frequency diagram at −15 dB has fewer noise points. It can effectively recover the emitter signal from the low SNR to facilitate the subsequent classifier processing.

The cyclic mean is set to:(20)$$\alpha =\frac{2\pi }{k}$$

The value of the cyclic frequency can be flexibly adjusted according to different needs. K is an arbitrary integer, the length of the cyclic sequence is proportional to the value of k, and the resolution is inversely proportional.

Figure 8 shows a diagram of the sequence length and domain transformation of the preprocessing method. Time series of the same length can be transformed into a one-dimensional sequence of cyclic domains of different lengths by calculating the mean value of different cyclic frequencies. The cyclic sequences are converted into time-domain sequences of the same length by the IFFT algorithm. At this time, the signal noise of multiple radiation sources received by the system has been suppressed. Since the number of size points of the CWD time-frequency analysis graph is the square of the length of the IFFT sequence, a short IFFT sequence length can significantly reduce the computation time of the CWD time-frequency image.

After preprocessing, the time-frequency images of the multi-radiation source signals are obtained, which can be directly fed into the classifier for feature identification. The pseudocode of the preprocessing algorithm is presented in Appendix A.

### 2.4. Computational Complexity of Preprocessing Algorithm

This paper’s preprocessing computational complexity analysis consists of three parts: cyclic mean, IFFT calculation, and CWD time-frequency analysis. The butterfly calculation method is adopted for the cyclic mean and IFFT, and four multiplications and two additions are defined as one operation. For the cyclic mean of N points, there are $N+\frac{N^{2}}{2}$ operations. Similarly, for an IFFT of k points, there are $k+\frac{k^{2}}{2}$ operations. The calculation amount of CWD time-frequency analysis is:(21)$$O(CWD)=\frac{\sigma k^{2}}{4}$$

In Formula (21), σ is a constant, t is the number of time points, and the computational complexity of the CWD algorithm is mainly related to t. If σ is 0.5, the CWD calculation of converting the cyclic mean series of 2000 points into an IFFT time series of 1000 points requires 657,750 calculations in the preprocessing process. If a 50 MHz embedded processor is used, the above steps take about 13 ms.

The computing speed and complexity in the natural environment depend on the hardware platform, the specific implementation of the algorithm, and many other conditions, so this paper calculates the running time on a microprocessor based on this theoretical estimation.

## 3. Multi-Emitter Signal Classification Model Based on a YOLO Deep Network

As a target-recognition algorithm in the field of deep learning, You Only Look Once (YOLO) has a high accuracy rate and short training time for multicategory object recognition. It is widely used in fault recognition, dangerous goods recognition, intelligent driving, and other fields that are characterized by clear returned images, obvious target features, and distinguishable features. Benefiting from the rapid identification of the YOLO algorithm for good multiobjective support, applying a small sample dataset to real-time signal sorting under low-SNR recognition technology has a great advantage, but often the subduction of the electromagnetic signal in the noise and the characteristic of time-frequency domain overlapping confusion cause the YOLO deep network to have low sorting result accuracy. However, after the multi-emitter signal preprocessing steps proposed in this paper, the problem of obtaining high-quality images is solved, and a high-resolution signal time-frequency map containing signal-feature information suitable for the YOLO algorithm is obtained, which is convenient for sorting and recognition operations.

The YOLO algorithm is based on a convolutional neural network architecture, which can simultaneously mark the features and locations of multiple targets. According to the architecture size of the network, it can be divided into five sizes: N, S, M, L, and X. Each size is composed of four parts: input end, backbone, neck and head (prediction). In general, the YOLO algorithm consists of the following steps:
An image is segmented into an S×S image matrix grid.If the center point of the predicted target falls within a grid, then the grid is bound to the current target.Within each grid, K borders containing the center point of the target are predicted.The target confidence of the border within each grid and the probability of each border area on multiple categories are calculated.The target window with low confidence is removed according to the threshold set.Non-maximum rejection (NMS) is used to remove redundant windows.

In the input-processing stage, YOLOv5 uses mosaic data enhancement, adaptive anchoring, image scaling, and other methods to preprocess images and splice multiple images in different ways to increase the number of features and improve the recognition accuracy of datasets with small sample sizes.

The backbone uses focus downsampling, a CSP structure, and an SPP pooling layer to extract image features. The neck adopts an FPN+PAN feature pyramid structure as the feature-aggregation layer of the entire network architecture, which can obtain the multiscale information of images and better solve the multiscale recognition problem of image features after the input to the head. The head uses three loss functions to calculate the feature classification, feature location, and confidence loss, and improves the accuracy of network prediction through NMS. The output of the algorithm includes a vector of prediction box category, confidence, and coordinate position.

The CBL structure of YOLOv5 uses a convolution (Conv) layer and a batch normalization (BN) layer and is passed into a SiLU activation function layer. The SiLU activation function is defined as follows:(22)SiLU(x)=x⋅σ(x)
(23)$$\sigma (z)={\left(1+\exp(-z)\right)}^{-1}$$

The focus module divides the input images into four layers, splices them into 12-dimensional feature images, extracts 3 × 3 feature information, and generates 32-dimensional feature images. Focus downsampling does not lose image information, significantly reduces the amount of computation, and improves the overall training speed.

The bottleneck consists of residual modules. It controls the number of channels using two Conv modules, extracts feature information, and controls residual connections using shortcuts.

The Conv module in SPP reduces the input number of the feature map, subsamples the maximum pooling of three different convolution kernels, and splices the results with the input feature map according to the channel. YOLOv5 uses convolution kernels with consistent step sizes to realize SPP, obtaining the same feature-map size but with different regional sensitivities, and finally realizing feature fusion.

Through the above structure, three steps of image scaling, convolutional neural network and non-maximum value suppression of object detection are realized.

The loss function of YOLOv5 includes three parts: localization loss, classification loss, and confidence loss, where the total loss function is the sum of the three-part loss function. The GIOU function for the confidence loss is calculated as:(24)$$GIOU=IOU-\frac{C-(A\cup B)}{C}$$
(25)$$IOU=\frac{(A\cap B)}{(A\cup B)}$$
where *A* is the prediction box of the model, *B* is the real box, and *C* is the minimum box containing *A* and *B*. For localization loss and classification loss, the YOLOv5 model adopts the cross-entropy loss function:(26)$$C=-\frac{1}{n}\sum_{x}\left[y\ln a+(1-y)\ln(1-a)\right]$$
where *x* is the sample, *y* is the label, a is the predicted value, and *n* is the total number of samples. The pseudocode for the Yolo algorithm has been presented in Appendix A.

## 4. Research on Multi-Emitter Signal Sorting and Recognition Based on the Preprocessing Method and the YOLOv5 Model

### 4.1. Experimental Environment and Dataset Setup

The experimental platform environment built in this study is shown in Table 1. An NVIDIA Tesla V100 graphics card and CUDA11.1.1 hardware acceleration greatly improve the training speed.

In this study, seven modulation categories and mixed categories of radiation-source signals are used. The single modulation category includes triangle wave linear frequency modulation (FM_TRI), sawtooth wave frequency modulation (FM_SAW), sinusoidal frequency modulation (SFM), AM, Fre_Agility signal (Fre_Agility), pulse signal, and biphase coding signal. Since this study studies the modulation-type recognition of radiation-source signals with similar frequencies, the recognition rate of single signal features will be greatly affected when the frequencies are similar or partially aliased. To improve the recognition rate of mixed signals with highly similar modulation features, as shown in Figure 9, frequency overlapping features of 15 classes of mixed signals with similar frequencies, such as FM_TRI, FM_SAW, and SFM, are added in the process of dataset annotation, for a total of 22 classes.

In Table 2, the parameters of the signal of multiple radiation sources are listed. The carrier frequency is set to 800 MHz–1.5 GHz, and the modulation center frequency is 100 kHz. To simulate randomness, signals are randomly generated within 10 kHz of the center frequency of each radiation source signal. The characteristic size of each signal in the time-frequency image is uncertain; for example, the triangle wave linear frequency modulation signal and sinusoidal frequency modulation signal have obvious characteristics in the case of a large bandwidth and are close to the pulse signal, binomial coding signal, and variable speed signal in the case of small bandwidth and modulation slope. The input to YOLOv5 uses Mosaic to optimize the effect of small target-feature detection. In this study, 3500 mixed-signal CWD image datasets are used for classification training tasks. In the process of data enhancement, the datasets can be further expanded by random cutting and stitching of the program to increase the number of features at the multiscale level.

Figure 10 shows the YOLOv5s data-enhancement result. Four time-frequency images are cropped, scaled, and then randomly arranged into one image to improve the generalization ability of the model. To better distinguish the states of signals in the frequency-domain overlap, this paper considers the law of signal overlap in the labeling of datasets and uses different label sizes to cover multisignal features under different parameters, so that the classifier can better learn the types of signals under the overlap.

As shown in Figure 11, the dataset contains single category labels and mixed category labels. Since the training dataset requires a large amount of manual labeling, the unlabeled dataset is also added to this study to participate in the training. The superposition mode of multiple signals is used as the label, and only the signal types need to be labeled on the whole image. The advantages of this method are as follows: first, a small number of unlabeled data images can be added to the labeled dataset for training, and the robustness of the classifier can be improved by adjusting the number of unlabeled datasets. Second, taking the unlabeled dataset as all the training data is more in line with the application scenario, can also reduce the tag-traversal time of the model in training in terms of the number of tags and improve the response speed of the system.

### 4.2. Experimental Analysis Based on the YOLOv5 Classification Model

Figure 12a shows the n size model, (b) the s size model, (c) the m size model, and (d) the x size model, which can be seen from the chart. The x in a CWD image-size model identifies the characteristics of the largest number, and the residual quantity is small, but the training of approximately 3000 copies of the image dataset takes a long time—here, it reached 50 h. The n-size model only takes approximately 6 h to train, but the missed detection rate and recognition accuracy cannot reach the classification standard. The m size model has a low recognition rate for small target features, so the s size model is selected for training in this study to achieve a balance between training time and comprehensive recognition effect.

To ensure the real-time performance of engineering applications, the boundary conditions of recognition accuracy and training time should be considered comprehensively according to the working environment. Therefore, this paper lists the training duration and recognition accuracy of YOLOv5 models of different sizes. As shown in Table 3, the dataset used in each experiment is the same as the experimental environment. It can be seen that the training time of YOLOv5n is the shortest, 376 min. The training time of the YOLOv5s model is 1170 min, and the ratio of training time to prediction accuracy is second only to YOLOv5x. Although the x size model has high accuracy, the training time is as high as 3000 min, which is not conducive to updating the dataset.

To facilitate the selection of the hardware platform, the GFlops value is listed in this paper. It can be seen that the computing pressure caused by different size models on GPU varies greatly. In a processing platform that needs a fast response, a suitable device can be selected quickly according to the real-time processing capability of the hardware processing platform.

To compare the recognition rate (PSR) of the signal preprocessing method proposed in this paper with other time-frequency analysis methods, −15 −0 dB time-frequency analysis images are used as input to the YOLOv5s model to obtain recognition accuracy. As shown in Figure 13, the trained model can distinguish subsignals of a single category and compound category. Due to the large improvement in the noise-suppression performance of the preprocessing algorithm, the characteristics of each signal can be highlighted at the image level, which improves the model recognition rate and reduces the probability of missing detection.

Figure 14a lists the recognition-rate accuracy of the mainstream time-frequency processing method and the preprocessing method proposed in this paper for mixed signals under 0–15 dB. After the preprocessing method, it is imported into the YOLOv5s model for feature recognition. Compared with common time-frequency analysis methods such as CWD, STFT, WVD, and wavelet transform, the data-preprocessing method proposed in this paper has certain advantages in anti-noise, so it obtains higher recognition accuracy at low SNRs. At 0 dB, the mainstream preprocessing method can maintain high recognition accuracy under the YOLOv5s model because of the small noise influence. It also proves the effectiveness of the YOLOv5s model.

Figure 14b shows the recognition-accuracy comparison between the proposed multisignal recognition framework and other LPI-NET frameworks. It can be seen from the figure that the CWD time-frequency analysis method has more advantages than STFT, WVD, and other methods in the multisignal preprocessing stage with a low SNR because CWD has better cross-term suppression performance. After using CWD time-frequency analysis, the network detection accuracy is more than 90% when approaching −8 dB, and the recognition accuracy is more than 50% when approaching −15 dB.

Figure 15a shows the comparison of recognition accuracy between this paper and other classification frameworks. As can be seen from the figure, under low SNR, the proposed method has higher recognition accuracy than other frameworks, and there is no phenomenon of declining accuracy. This paper conducts a 10-fold cross-validation experiment to comprehensively measure the recognition accuracy and stability of the YOLOv5S model. It can be seen from Figure 15b that the recognition accuracy of different sub-datasets fluctuates to a certain extent, but the recognition accuracy is 53%.

This paper compares the training time and mAP value of the YOLOv5s model and other classifiers. As can be seen from Figure 16, the training time of the YOLOv5s model is 19.5 h, about three times that of the n-size model, but the mAP value exceeds most classifiers. The YOLO5s model has a faster training speed and higher recognition accuracy. Although it does not exceed the x size model, the accuracy of the s size model is acceptable compared to the ultra-long training time. If the calculation time is prolonged, the precision can be improved again from the aspect of preprocessing.

Table 4 lists the single classification time consumption for different sizes of YOLO models and other classifiers. The single classification time consumption of the YOLO model is not directly related to size. Compared with the single recognition time of other classifiers, the single recognition time of the YOLO model is significantly improved compared with other classifiers, which has significant advantages in real-time performance.

To evaluate the stability and convergence effect of the training process, this paper lists the curve of the loss function and mAP value. Figure 17 shows the convergence process of the loss function and the change curve of the mAP value of the YOLO5s model in the training process. In this paper, 1000 cycles are used to complete the training process, and the loss curve tends to converge when approaching 1000 cycles without any mutation. The mAP value is continuously improved to achieve the ideal training effect.

## 5. Conclusions

Aiming at the identification problem of radiation source modulated signals in a low SNR environment, this paper proposes a signal preprocessing method based on the fusion of cyclic mean denoising, IFFT, and CWD time-frequency analysis. The length of the cyclic domain is adjusted by controlling the cyclic frequency. This can not only ensure the noise-suppression effect but also reduce the calculation amount of CWD time-frequency analysis and improve the real-time performance of the system. The noise-suppression effect is obvious when the SNR is above −15 dB. When it is lower than −15 dB, the operation time is significantly increased to ensure the noise-suppression effect.

The time-frequency images were identified using the YOLOv5s model, and a radiation source mixed signal dataset composed of 3500 CWD time-frequency images was generated. The classifier was trained by two forms of fusion, labeled and unlabeled. The classifier was able to identify independent signals and mixed-signal features and mark the position of signal features in the graph, paving the way for further parameter estimation. Compared with the single CWD, STFT, WVD, and wavelet transform time-frequency analysis methods, the proposed preprocessing method performs better in the classifier and is less affected by SNR fluctuations. The prediction accuracy with SNR changes has no mutation or decline, and the signal-classification accuracy under low SNR increases significantly. In the process of training, the loss function and mAP value mutations are lower, and an ideal training effect is achieved. In the process of signal preprocessing, the problem of low accuracy of the YOLO network for signal-feature recognition under a low SNR is solved, and the advantages of the preprocessing algorithm and classifier can be organically combined.

## 6. Future Work Outlook

The real-time signal processing scheme proposed in this paper can be applied to embedded systems, and we will further improve the preprocessing and classifier work in the future.

Firstly, we will port the technical framework to the deep learning framework.

Secondly, we will use the output results of the classifiers on the time-frequency graph to estimate the signal parameters of the mixed emitter. This estimation method does not need to be calculated by the model. It just needs to read the size and position of the marker box in real-time.

Finally, we will use the signals collected in the real world to generate large-scale mixed-emitter signal datasets in the form of manual and automatic labeling. After the production of the dataset, we will transplant the technical framework into a high-performance embedded system with a deep learning function, which can complete online learning, real-time analysis, and database updates.

## Figures and Tables

**Figure 1 sensors-22-07783-f001:**
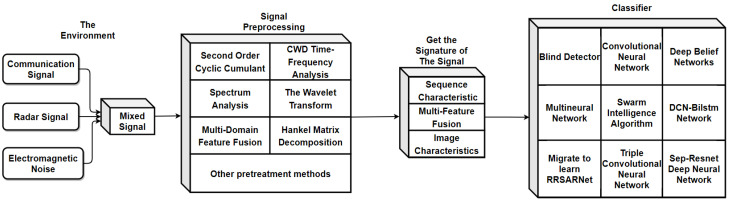
Roadmap of mainstream radiation source signal-sorting and identification technology under a low SNR.

**Figure 2 sensors-22-07783-f002:**
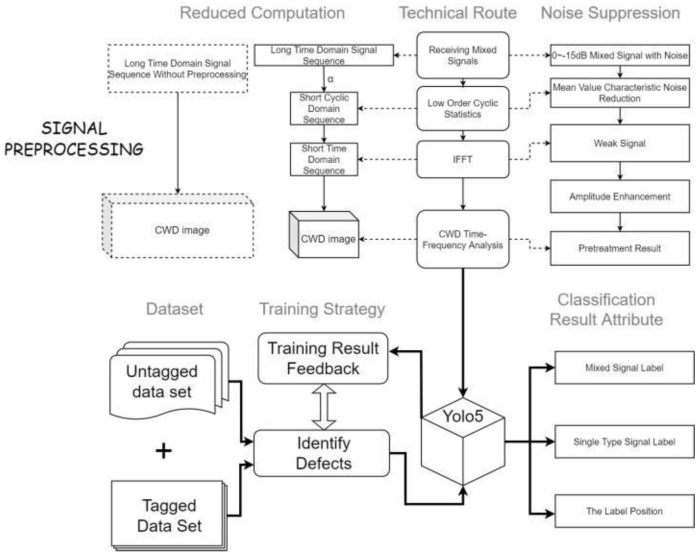
The technical process of this paper.

**Figure 3 sensors-22-07783-f003:**
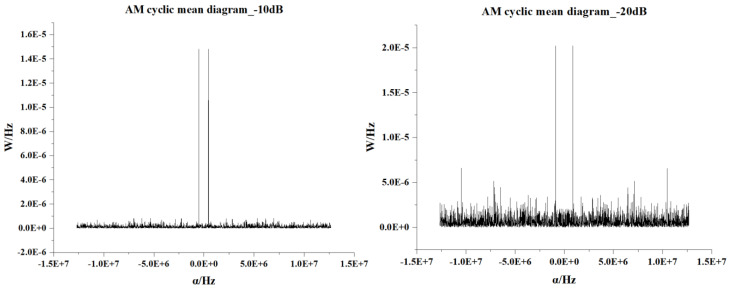
AM signal cyclic mean at different SNRs.

**Figure 4 sensors-22-07783-f004:**
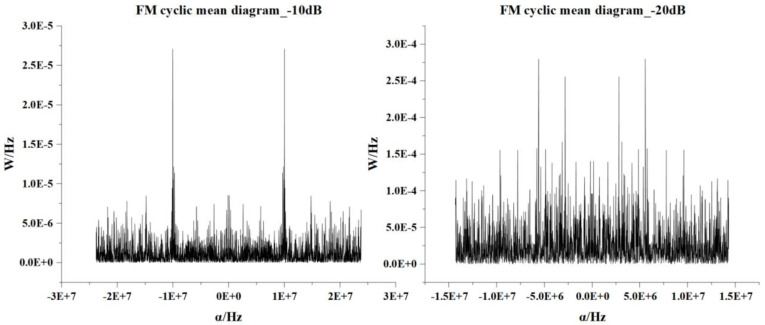
The cyclic mean of the FM signal under different SNRs.

**Figure 5 sensors-22-07783-f005:**
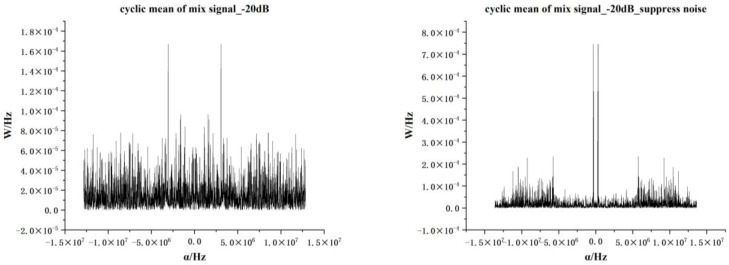
The cyclic mean of the AM and FM multiple signals.

**Figure 6 sensors-22-07783-f006:**
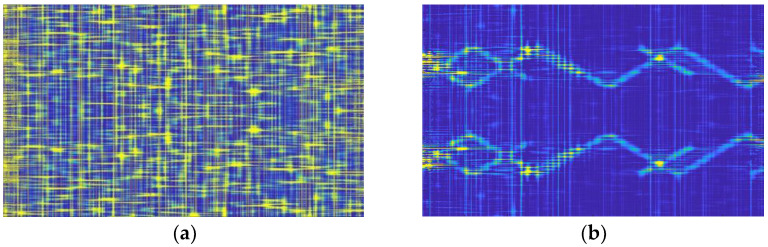
(**a**) unsuppressed noise (−20 dB), (**b**) suppressed noise.

**Figure 7 sensors-22-07783-f007:**
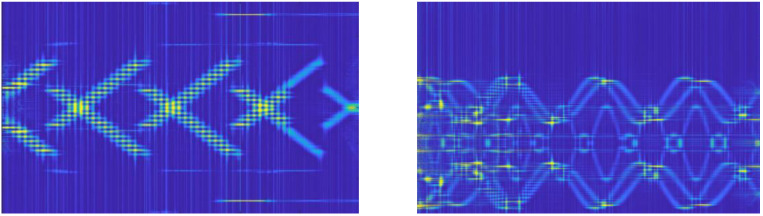
Time-frequency diagram of multisignal CWD after noise suppression (−15 dB).

**Figure 8 sensors-22-07783-f008:**
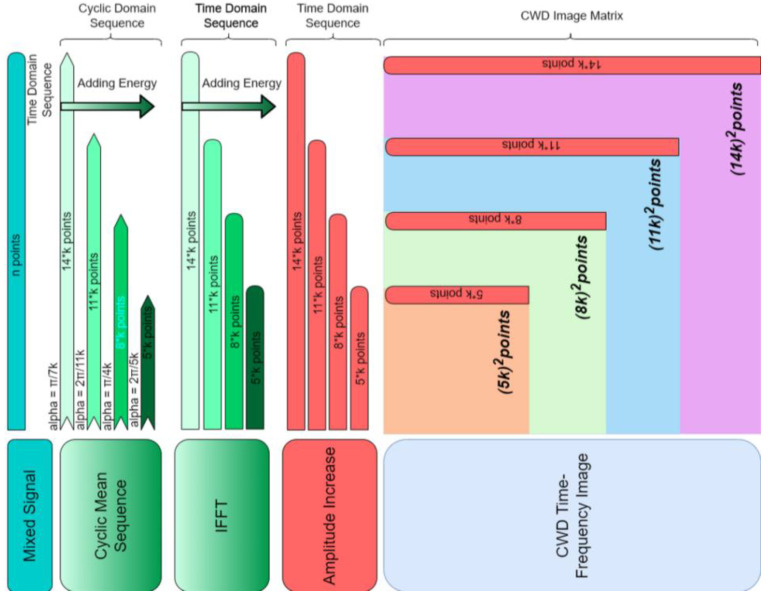
Preprocessing method steps and sequence length comparison diagram.

**Figure 9 sensors-22-07783-f009:**
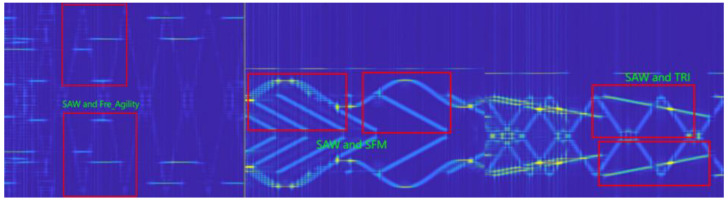
Mixed feature markers.

**Figure 10 sensors-22-07783-f010:**
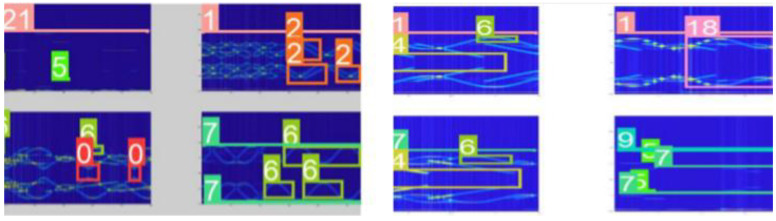
Data enhancement of the CWD time-frequency graph training process by input.

**Figure 11 sensors-22-07783-f011:**
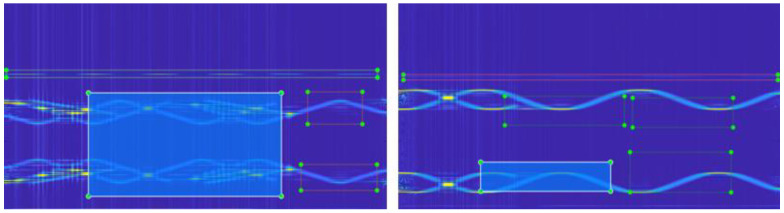
Range of labels in the case of multiple signal overlap.

**Figure 12 sensors-22-07783-f012:**
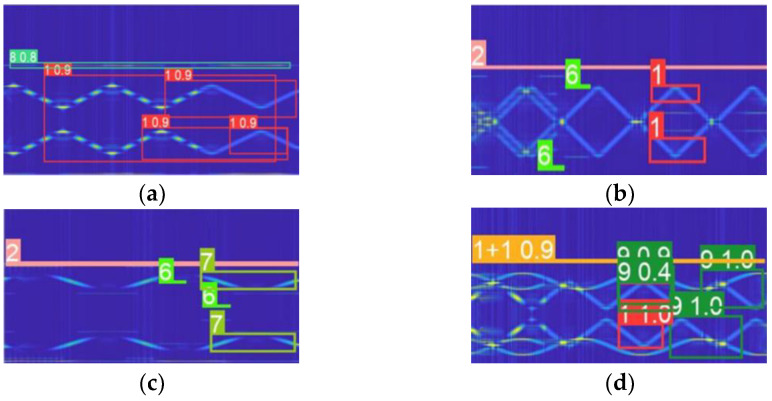
Training validation procedure of the four models.

**Figure 13 sensors-22-07783-f013:**
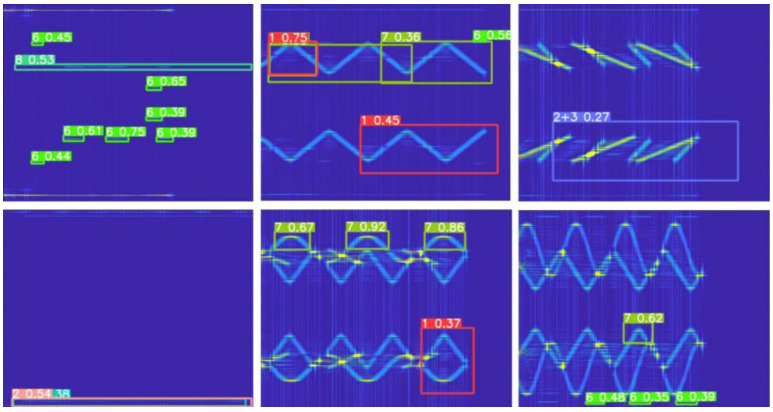
Identification results of the YOLOv5s model.

**Figure 14 sensors-22-07783-f014:**
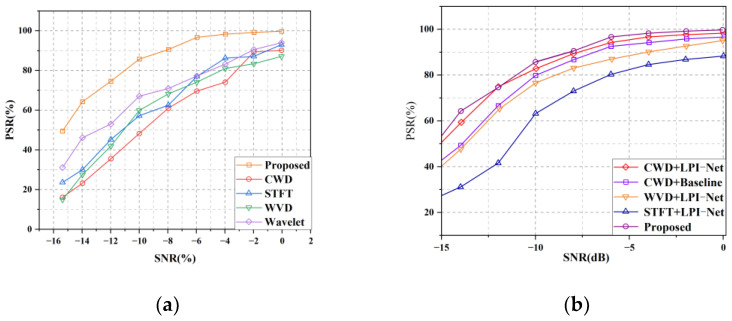
Recognition accuracy of different preprocessing methods in YOLOv5s: (**a**) the recognition accuracy of different preprocessing methods in YOLOv5s and (**b**) the comparison of recognition accuracy between the proposed multisignal recognition framework and LPI-NET and other frameworks.

**Figure 15 sensors-22-07783-f015:**
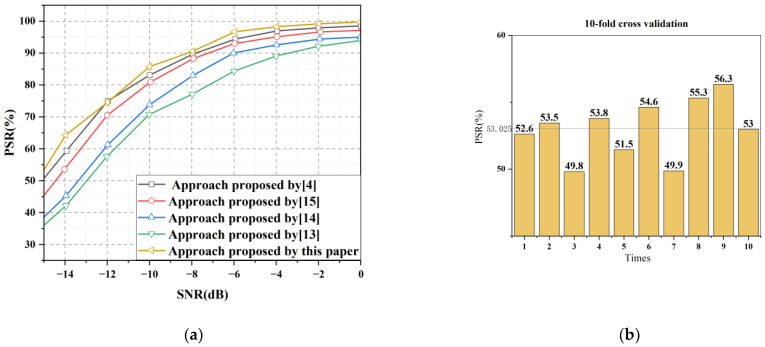
(**a**)comparison between the proposed multisignal recognition framework and CNNs in other papers. (**b**) 10-fold cross-validation of −15 dB mixed signals.

**Figure 16 sensors-22-07783-f016:**
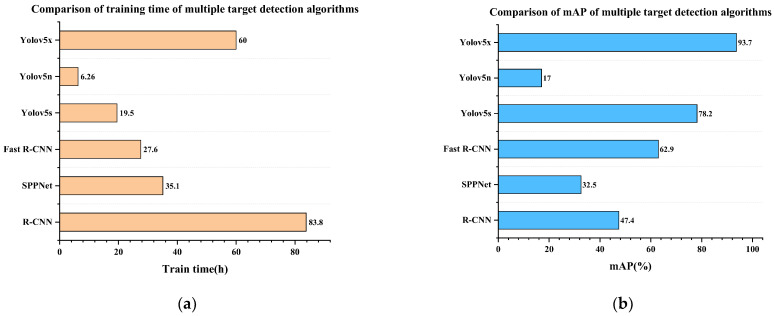
(**a**) Training time of different object detection models. (**b**) mAP values of different object detection models.

**Figure 17 sensors-22-07783-f017:**
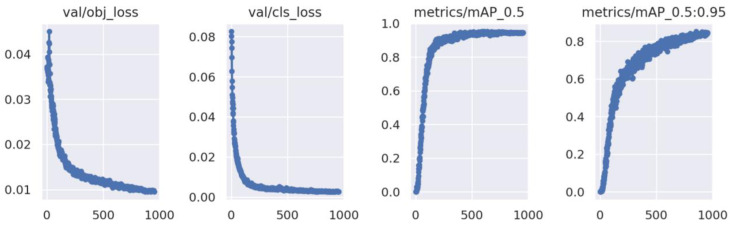
Loss curves with mAP.

**Table 1 sensors-22-07783-t001:** Experimental platform environment configuration.

Category	Configuration
CPU	Intel(R) Core(TM)i7-10700K @3.80 GHz
GPU	NVIDIA Tesla V100 32 GB
Operating System	Linux
Deep Learning Framework	Pytorch 1.8.1
Programming Language	Python 3.8
Dependent Package	CUDA 11.1.1

**Table 2 sensors-22-07783-t002:** Experiment parameters (Single Classes).

	Carrier Fre	Mod Fre	Sample Fre	SNR	Special PAR
FM_TRI	1 GHz	100 kHz	4 GHz	−15~0 Step	None
FM_SAW	1 GHz	100 kHz	4 GHz	−15~0 Step	None
SFM	1 GHz	100 kHz	4 GHz	−15~0 Step	None
PULSE	2 GH z	100 kHz	4 GHz	−15~0 Step	Duty Ratio 50%
AM	800 M–1.5 GHz	100 kHz	4 GHz	−15~0 Step	50% Mod Rate
Fre_Agility	800 M–1.5 MHz	100 kHz	4 GHz	−15~0 Step	5 Frequencies
Biphase Coding	1 GHz	100 kHz	4 GHz	−15~0 Step	Duty Ratio 50%

**Table 3 sensors-22-07783-t003:** Training results of models with different sizes (−15 dB mixed radiation source signal).

Size	Precision (%)	GFLOPs	Parameters	Data Pictures	Training Time (min)
**YOLOv5n**	42.3%	6.0	2.9 M	3500	376
**YOLOv5s**	53%	17.1	7.7 M	3500	1170
**YOLOv5m**	89.2%	43.6	21.4 M	3500	1982
**YOLOv5x**	96.5%	219.7	219.0 M	3500	3000

**Table 4 sensors-22-07783-t004:** Test time of multiple target-detection models.

YOLOv5n	YOLOv5s	YOLOv5x	R-CNN	SPPNet	Fast-R-CNN
0.37 s	0.4 s	0.45 s	47 s	2.3 s	0.32 s

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
