# Peer review of "Low SNR Multi-Emitter Signal Sorting and Recognition Method Based on Low-Order Cyclic Statistics CWD Time-Frequency Images and the YOLOv5 Deep Learning Model"

_sensors, 2022, doi:10.3390/s22207783_

Round 1

Reviewer 1 Report

In the paper titled "Low SNR Multimitter Signal Sorting and Recognition Method   Based on Low-order Cyclic Statistics CWD Time-frequency Im- 3  ages and the YOLOv5 Deep Learning Model" authors proposed a multiemitter signal feature sorting and recognition method based on low-order cyclic statistics CWD time-frequency images and  the YOLOv5 deep network model that dissociates, label, and sorts the multiemitter signal features in the time-frequency domain under a low SNR environment.

My few suggestion to improve the manuscript are as follows.

1. Although at the end of Section 1, authors listed the main innovations of this manuscript. However, to make it more clear and easy to understand, I suggest authors should mention their contributions in the bullets form. This will essentially make the contributions much clear for the readers.

2. Authors have used so  many acronyms in the manuscript. I suggest at the end of Section, include a acronym table and their meanings that are used through out the paper. Later then those acronyms can be used.

3. In Section 2 and Section 3, authors have messed up many equations and mathematical expressions. I suggest to improve the write up. One simple suggestion is for your proposed method include a pseudo code that clearly indicates the flow of the algorithm and all the parameters used therein.

4. For a journal manuscript, this manuscript contains very limited References. I suggest to include more recent references, especially focusing on year 2022.

5. In section 5, authors should add few sentences about the possible future work. Currently, only conclusions is listed. 

6. Language used in this manuscript needs attention. At few place, full stops and spacing are also desired. For instance, in abstract, after ... SNR environment dot, a space should be inserted. Similarly, authors should look for such minor issues through out the manuscript.

7. In Figure 2, all the blocks should be made transparent. In print version, these are not clearly visible. Moreover, for this figure, a pseudo code should also be included.

8. After equation (13), alignment of parameters used should be rectified. Currently, it just floats in the air. Similarly, authors should rectify all such issues related to the equations parameters description.

9. In Section 3, a pseudo code about YOLO deep network should be included. Current, description does not provide the readers the full insight.

10. I would suggest authors to include the computational complexity of their proposed algorithm. 

Author Response

Your comments have made great progress in our work. On behalf of the author team, I would like to thank you for your efforts!

Reviewer 2 Report

1) Does the paper contribute to the body of knowledge?

This paper addresses the challenge of sorting and recognising multimeter signals with low SNR in this paper. The authors first preprocess the signal using cyclic mean and CWD time-frequency analysis to tackle the problem. The preprocessing stage suppresses noise in the signal and acquires features needed for the next step – signal recognition. The signal recognition is done using the YOLOv5 network. The YOLOv5 network recognises different signals in the time-frequency images of the multi-radiation source signals. In the experimental part, the authors generate synthetic signals with variable shapes and SNR to discover the limitations of the proposed method. Authors analyse the performance of four YOLOv5 architectures, each with a different computation-accuracy trade-off. In addition, the authors examine the choice of preprocessing stage and classifier by comparing them with existing approaches (e.g. STFT, SIFT and others for preprocessing step and LPI-NET for the classifier). Comparison with existing state-of-the-art work reveals that the proposed solution offers improvements in signal recognition for lower values of SNR. The paper undoubtedly contributes to the body of knowledge. The presented results justify the method employed. Issues:

  1. Although improvements in detection can be seen, Figure 15 reveals that improvements are low. We can see that accuracy increases by only 5% for -15 dB, while for the rest of the cases, the method in [15] is equally applicable. Please perform 5- or 10-fold cross-validation to ensure statistical correctness of results. 
  2. The authors employ synthetically generated data to train the model and evaluate the method, representing my biggest concern. Is it possible to assess the model on real-world data? Please consider publishing your data.
  3. In Table 3, the authors discuss the training duration of different YOLO networks. I fail to understand why the training duration is necessary and how it influences the network choice. Please address that. Furthermore, what does GFLOPS in Table 3 mean? Are these GFLOPS needed for training or inference? 
  4. The choice for preprocessing step and YOLO network is justified with computation complexity. Do authors envision an embedded system or similar solution for on-site signal recognition, or will the method be part of a pipeline executed by a computationally powerful processing unit? 

2) Is the paper technically sound?

a)     Grammar is the biggest issue: sentences are hard to read. Although the article poses the potential for publishing in the submitted journal, it needs to go through extensive spell checking. I would suggest Grammarly or InstaText for spell-checking.

b)     The Introduction should be rewritten. It is hard to read (especially the first paragraph), and the reader fails to understand the bigger picture, the importance of signal recognition, etc.

c)     The abbreviations need to be defined. I recommend that the authors add a special section in the paper to explain abbreviations.

3) Is the subject matter presented in a comprehensive manner?

a)     Section 3.2 needs attention. Authors are jumping between figures without giving the motivation behind pursued evaluations (Results for Figures 16 and 17 and table 4).

b)   Equations in 2.1 should be part of a separate appendix

Author Response

(The authors gave the same response as above.)

Round 2

Reviewer 2 Report

The authors have put a great effort into improving the paper. They have addressed all my comments and suggestions. I recommend that the article be accepted for publication.

Minor concerns:

  1. Add section 2.1.2 Cyclic mean analysis of FM signals. It would be best if you separated the analysis of FM and AM signals.
  2. Lines: 319 - 320. The execution time on the microcontroller is a theoretical assessment. Please state that in the article. 
